# Understanding the Experience and Needs of School Counsellors When Working with Young People Who Engage in Self-Harm

**DOI:** 10.3390/ijerph16234844

**Published:** 2019-12-02

**Authors:** Ben Te Maro, Sasha Cuthbert, Mia Sofo, Kahn Tasker, Linda Bowden, Liesje Donkin, Sarah E. Hetrick

**Affiliations:** 1Department of Psychological Medicine, Faculty of Medical and Health Sciences, University of Auckland, Private Bag 92019, Auckland 1142, New Zealand; ben.te.maro@auckland.ac.nz (B.T.M.); l.bowden@auckland.ac.nz (L.B.); l.donkin@auckland.ac.nz (L.D.); 2Department of Psychology, University of Auckland, Private Bag 92019, Auckland 1142, New Zealand; scut898@aucklanduni.ac.nz (S.C.); ktas226@aucklanduni.ac.nz (K.T.); 3Orygen The National Centre of Excellence in Youth Mental Health, 35 Poplar Rd, Melbourne, Victoria 3052, Australia; 4Centre for Youth Mental Health, University of Melbourne, Melbourne 3010, Australia

**Keywords:** self-harm, suicide, school, guidelines

## Abstract

Self-harm rates are increasing globally and demand for supporting, treating and managing young people who engage in self-harm often falls to schools. Yet the approach taken by schools varies. This study aimed to explore the experience of school staff managing self-harm, and to obtain their views on the use of guidelines in their work. Twenty-six pastoral care staff from New Zealand were interviewed. Interviews were analyzed and coded using thematic analysis. Three themes emerged: The burden of the role; discrepancies in expectations, training, and experience; and the need for guidelines to support their work. This research, therefore, demonstrated a need for guidelines to support school staff to provide support around decision making and response to self-harm in the school environment.

## 1. Introduction

Worldwide, rates of youth self-harm (self-inflicted injury of self-poisoning irrespective of motivation or the individual’s intent to die or not [1]) are increasing [2]. Whether or not the distress that drives self-harm includes an explicit wish to end one’s life, youth who engage in self-harm are at an increased risk of suicide [3,4] and a range of potential long-term negative consequences [5,6,7,8].

Within New Zealand the rate of self-harm [9] parallels the worldwide trend and the rate of youth suicide is the highest of any Organization for Economic Co-operation and Development (OECD) country [10]. Depending on how it is measured, between one quarter and close to a half of secondary school students in New Zealand have engaged in self-harm, and up to 21% have seriously thought about suicide in the previous 12 months [9,11,12]. Clustering of self-harm in young people is of concern given youth suicides are twice as likely to occur in clusters than adult suicides [13], and because young people who have been exposed to self-harm among friends and/or whānau (family) are at increased risk of self-harm [14].

Young people spend much of their time in the school environment and it is a key influencer of psychosocial and emotional development [15,16,17,18,19,20]. Given this, it is an important context for prevention and intervention in self-harm. Moreover, while young people can be reluctant to seek professional help [21,22], they may be more comfortable with the school counsellor compared with other health professionals [23]. It is also the case the that schools with increased availability of mental health services report fewer depressive episodes, suicidal ideation, and suicide attempts in their students [24,25].

Schools are a logical and accepted setting for suicide prevention activities, and there is growing evidence to support this [26,27]. It is also critical that schools are swift and effective in their management of students who engage in self-harm in order to support the young person and ensure the risk of contagion is minimized. School staff, despite often being the first port of call for at-risk students, report feeling limited in their capacity to respond appropriately to these students [20,28,29]. There is a lack of clarity around how well schools and their staff, including those specifically responsible for the mental health and wellbeing of students, are resourced and equipped to engage in this kind of work. This can sometimes result in confusion about how to respond, unhelpful responses to self-harm [30], and a lack of clear plan to support the student and the school [20].

Recently, there has been a focus on supporting staff and students through position papers and guidelines outlining when and how to respond to self-harm in the school environment [30,31,32]. However, these do not adequately address the unique cultural challenge of working in New Zealand. The aim of this project was to understand the perspective of the key school personnel in New Zealand responsible for the mental health and wellbeing of students. We wanted to know what their experience was of managing students who engaged in self-harm. Gain insight into how these students were managed within the school, including the use of guidelines, protocols, and procedures, as well as understand how well-resourced staff were to manage these studies. Finally, we sought to understand if guidelines were required, or would be welcomed.

## 2. Methods

### 2.1. Study Design

This was a thematic qualitative study with data collection taking place via individual interviews, guided by a semi-structure interview schedule. A general inductive approach was used, such that themes were derived in the context of specific research aims and objectives [33].

### 2.2. Participants

Twenty-six pastoral care providers from 16 secondary schools in New Zealand were interviewed (see Table 1 for participant characteristics). Purposive sampling was utilized [34] to ensure representation from state, state integrated, and private schools, encompassing all deciles from one to ten (a key measure of socioeconomic status), and geographical locations across Auckland, the largest city in New Zealand, and two schools from outside of Auckland. While research suggests that meta-themes typically emerge within six to twelve interviews [35] the current study conducted 26 interviews in order to include this diversity of schools.

Four schools were decile ten, two were decile nine, two were decile eight, one was decile seven, one was decile five, three were decile four, one was decile three, one was decile two and one was decile one (a lower decile school has more students from lower socio-economic communities). Nine schools were state-funded schools, three were state-integrated schools (former private schools that have been integrated into the state system), and four were private schools, with two of these being boarding schools. Eight schools were in central Auckland, two were in south Auckland, one in west Auckland, one in the east Auckland, and three in the north of Auckland. There was one additional school outside of Auckland.

The pastoral care providers were mostly trained counsellors, with one who was a Chaplin and one who was a social worker. Of the pastoral care providers 21 identified as female and seven as male. The majority identified as New Zealand European, one identified as Dutch, one as Samoan, one as Asian, and one as English.

### 2.3. Procedure

Emails were sent to the researchers’ existing networks with a request to contact the researchers by email or phone if participants were interested in participating. Researchers then met with the pastoral care staff, at which time participant information was provided verbally and in written form, and written consent was obtained before beginning the interview.

Interviews, which followed an interview guide, took between 45–80 min. Interviews were audio recorded and transcribed verbatim. Participants were offered the opportunity to listen to their recordings or read their transcripts, allowing for edits or changes to be made; however, no participants chose to engage in this process.

All subjects gave their informed consent for inclusion before they participated in the study. The study was conducted in accordance with the Declaration of Helsinki, and the protocol was approved by the Southern Health and Disability Ethics Committee on 23rd of August 2018 (18/STH/30).

### 2.4. Analysis

Interview transcripts were analyzed using thematic analysis as outlined by Braun and Clarke [36]. This involved identifying, analyzing, and categorizing a pattern of themes within and across interviews [37]. Initial codes were generated; these were based on initial ideas that are relevant and consistent across participants. The coding was based on the research aims not research questions [34]. These codes were then organized into meaningful groups, which were iterated to form overarching themes.

A selection of interviews (*n* = 6) were double coded as the coding frame was being developed to ensure consistency, and this was discussed in supervision provided to ensure critique of the emerging themes [38]. Table 2 demonstrates the codes and how these were organized to form themes.

## 3. Results

Three themes emerged from the data: (1) burden driven by excessive workload and poor resourcing; (2) discrepancy, evidenced by differences in the training pastoral care staff received, the processes and procedures implemented and the approaches to communication both within and across schools; both of which contribute to the third theme: (3) the need for guidelines. Under each of these themes were several subthemes as highlighted in Figure 1 and Table 2.

### 3.1. Theme 1: Burden

Within this theme, two subthemes emerged that indicated an excessive workload and responsibility combined with under resourcing created a burden that made managing self-harm difficult. All participants spoke about the challenges of increased demand due to increasing numbers of students presenting with mental health issues, and a lack of support and resources to support participants in managing these students.

#### 3.1.1. Subtheme 1: Increased Mental Health Issues

Within this subtheme were two further but related categories. One was the increased prevalence of suicidal ideation, self-harm, anxiety, and depression in students, and that there has been a shift to the more severe and difficult to manage behaviors:


*“I’d say more suicide ideation and less self-harm at the moment anyway”*
*[Interview 5]*


*“I think definitely that self-harm has become more prevalent perhaps”*
*[Interview 18]*

It was also clear from the data that there was an increase in what pastoral care staff saw as potential drivers of mental health issues, including increased technology use and social media engagement. Both were believed to increase exposure to self-harm, increased information about how to engage in self-harm, decreased connection, and increased a sense of isolation. This behavior was also believed to lead to increased difficulties with emotion regulation, poorer coping skills to cope with stressful situations, and increased pressure on young people to conform and succeed to the “ideal norm” portrayed in these media:


*“And I do think that the level of anxiety and depression is really, um, increased because of societal pressure…”*
*[Interview 13]*


*“there will be a group of ten kids talking about some, something on social media about kids cutting themselves and how you do it”*
*[Interview 26]*


*“It’s just connections on their phone or whatever but no real friends”*
*[Interview 21]*

#### 3.1.2. Subtheme 2: Role Overload

Pastoral care staff described a change in role responsibility where they were taking on more and more students, and that students with self-harm were increasingly taking precedence over other types of presentation. Concurrently, there were few resources or support systems to support this, including not enough staff, for budget for things such as training:


*“There’s too many people coming through the door”*
*[Interview 11]*


*“Our biggest gripe here is that we’re understaffed”*
*[Interview 5]*


*“Someone improves and they’re okay, and then you have another 10 referrals waiting”*
*[Interview 32]*

Participants spoke about being required to be constantly available to support young people and therefore being at an increased risk of apathy or burnout. Many were concerned that their workday primarily consisted of managing young people engaging in self-harm, at the expense of seeing those deemed lower risk. There was a concern, that due to the lack of early intervention that initially lower risk young people’s behavior may escalate and their functioning may deteriorate.


*“I got a text message when I was out doing something else. Just like I’ve cut myself and I’ve taken a whole lot of pills”*
*[Interview 3]*


*“…well, sorry you don’t even get in my door unless there’s something much more specific going on”*
*[Interview 13]*

### 3.2. Theme 2: Discrepancy—Differences in the Way That Self-Harm is Managed

Within this theme, three subthemes emerged that indicated discrepancy across the quantity and type of training that pastoral care staff had received, the processes and procedures used in managing students who had engaged in self-harm and in communication channels and approaches within and external to the school.

#### 3.2.1. Subtheme 1: Training

This subtheme related to a general lack of initial professional training in how to manage self-harm, and considerable variation in how pastoral care staff then acquired these skills. Because of this variation, as well as variation in process and procedures (see subtheme 3.2.2), there was uncertainty and a resulting idiosyncratic and non-systematic approach to educating teachers and parents/whānau in how to support students.

Contributing to the variation in training is the diversity in school pastoral care staff. School pastoral care staff came from a range of disciplines, and there was little initial training once they entered the role. Few reported receiving training in how to manage young people who were self-harming or had suicidal ideation:


*“I don’t remember anything on self-harm”*
*[Interview 6]*


*“…not a single mention of it in 2 years training as a guidance counsellor”*
*[Interview 14]*


*“…none to very little, I can’t even remember. I would say it was none”*
*[Interview 20]*


*“I wouldn’t say we got any specific training in self-harm…. A lot of my training has come through external training from my [previous job]”*
*[Interview 31]*

Therefore, pastoral care staff relied on a range of ways to gain the knowledge and skills they felt they needed, including professional development, mentoring, advice from colleagues, and supervision. Some schools, often private schools, had more resources to allow staff to access increased training. While other schools were in areas where the local mental health services provided training:


*“I guess I just really relied on supervision and talking about it”*
*[Interview 3]*


*“…so, I watched her and I picked up what she said and what she did”*
*[Interview 20]*


*“There have been a couple of really good workshop seminar days on suicide risk assessment and that’s been very good, but then also I’ve just kind of done my own thing”*
*[Interview 20]*


*“I can go to a really good supervisor you know, but I know it’s all private and state schools are all challenged by budgets”*
*[Interview 26]*

There were idiosyncratic and non-systematic approaches that individual pastoral care staff had within a school for educating and supporting the wider staff. This meant further variation between the schools and in the training that staff within the same school received:


*“We do a presentation in the staff room to all staff around high risk and uh issues of depression and also suicidality or the possibility of suicidality”*
*[Interview 22]*


*“Meet with all the new stuff at the start of every year and explain what we do here and the services we have”*
*[Interview 20]*


*“we do keep a running roll of people that have been referred to us or we suspect are at risk of serious self-harm…and we gauge the risk level at low, medium, or high depending on who they are, whether they’re had a connection with someone who has taken their life, or if they have self-harmed in the past. Or their personal circumstances at home.”*
*[Interview 31]*

There was also a sense that training needed to be delivered to support parents who did not know how to manage a disclosure of self-harm or suicidal ideation from their child, or those that did not take the disclosure seriously such as due to cultural barriers or personal beliefs about these behaviors. Again, it appeared that there was an idiosyncratic and non-systematic approach to managing these situations. Parental involvement was also relevant to role overload (see subtheme 3.1.2). There was a general sense that there was a lack of resources and support to include parents in management plans within the school context:


*“I think parents just have no idea, they just have no idea…It’s all, it’s either we don’t want to think about this or we’re going to take complete control of it”*
*[Interview 10]*


*“If you actually allow skills around parenting….to communicate that to the children to self-regulate”*
*[Interview 22]*


*“I had a [ethnicity] parent last term, you know your child is suicidal, she tried to commit suicide last year mum says oh I just don’t believe”*
*[Interview 14]*


*“I don’t have capacity to work with the parents as well, yeah. So that’s, part of, a rock and a hard place with that at the moment. Seeing that yes, they need to be part of the conversation, don’t know how to do that well”*
*[Interview 10]*

#### 3.2.2. Subtheme 2 Process/Procedure

Responses revealed considerable uncertainty, and variation in practice in terms of risk assessment, management of confidentiality, referral to appropriate agencies, management and intervention strategies, and the general wider school approach to wellbeing and the prevention of self-harm. First there was variability in terms of who undertook the risk assessment, when it was completed, and how it was implemented including whether specific tools were used. There was a general tendency consistent across participants to categorize risk based on key questions about suicidal thinking. The participant responses revealed a desire for some consistency and guidance about what and how to ask across participants:


*“The deputy principal is somehow supposed to assess risk of which they’re not trained to assess”*
*[Interview 19]*


*“So, every young person gets exactly the same questions I think that’s really important. Can I say while this is being recorded it would be so good if nationwide everybody did the same assessment”*
*[Interview 20]*


*“I make a distinction in my own head, with the client, between self-harm and suicide risk”*
*[Interview 23]*


*“I might use a tool maybe an anxiety scale or a depression scale”*
*[Interview 26]*


*“The questionnaire then identifies a group of high-risk students, medium risk, low risk”*
*[Interview 12]*

The lack of consistency and confidence in formulating risk then leads to concern and uncertainty around confidentiality and if school pastoral care staff should break confidentiality and inform parents or other services. The decision to break confidentiality, whilst often based on risk, was perceived to be further complicated by considering the young person’s age and insight. Moreover, there was considerable variability across staff regarding what sort of behavior and thinking will be tolerated before confidentiality should be broken. There was considerable concern about the impact of breaches of confidentiality on the therapeutic relationship, with some examples of how this is mitigated:


*“…like, yeah, level of safety is an issue like in terms of whether confidentiality gets broken or not”*
*[Interview 26]*


*“So, there have been times when I will not disclose that to a family member if they’re over 16”*
*[Interview 15]*


*“So, my confidentiality is really high unless it’s immediate, imminent risk…Uh… having said that, seeing year seven and eight’s … I’m just like actually you’re so young that I’m going to have to have a conversation with them around actually we need your parents involved in this”*
*[Interview 10]*


*“We would be a little less willing to sit on disclosure around self-harm, though we do make a distinction between self-harm disclosure and suicidality”*
*[Interview 22]*


*“Confidentiality breaching that has to happen at times inevitably puts your relationship is at risk. Our relationship stays more protected if the nurse does the informing of family as an accompli without negotiating with the student”*
*[Interview 12]*


*“The kids get the confidentiality speech, so do the staff if they come in”*
*[Interview 23]*

The decision to refer the young person to another service was often tied to the risk assessment warranting breaching confidentiality. There were difficulties with referral when the young person preferred to remain supported by the school staff with whom they had an existing relationship rather than being referred elsewhere. There was variation in the willingness of the pastoral care staff to refer a student on to outside agencies, often influenced by how the perceived the quality of the relevant agency. The reluctance to appropriately refer was related to the variation and uncertainty of communication taking place with external agencies (see below 3.2.3 highlighting differences in how external agencies were seen). There were also idiosyncratic approaches to how referrals are made within schools and who they were made to:


*“If they’ve got a plan for suicide always refer on, if they don’t have a plan still might refer on depending on history and self-harm, but there is not always a need to refer. But in both cases always call home”*
*[Interview 20]*


*“We do a referral, um… sometimes it’s very difficult, um, to get it to stick because, um, the students have an ongoing relationship with us”*
*[Interview 5]*


*“We’ve got a referral form that the teachers use, so they have to fill that in and email that to me”*
*[Interview 20]*

The data also revealed differences regarding who would see the student, be involved in their ongoing management, who would be the primary treating person or be working in what role (this links to 3.2.3 highlighting the different ways that different professional staff work together). The data revealed some variability and confusion in terms of whether self-harm wounds were required to be medically dressed and there is substantial variation regarding what therapeutic approaches were used:


*“…cutting that needs addressing, you need to get the nurses involved and the nurses and you work collaboratively and if [student] presented to you and the trust’s there, you’re the lead so I’ve created the lead caregiver… a lead caregiver which is defined in this stage by relationship to child”*
*[Interview 1]*


*“But many of the nurses that come newer to our departments will engage in counselling. And where I sit, I think it’s very unhelpful….it needs to be done by a counsellor who has the skill …”*
*[Interview 12]*


*“Students had to have self-harm incidences or scratches or whatever covered…but I just read that isn’t the recommended process and procedures”*
*[Interview 18]*


*“If someone has presented that has been harming. The focus is on what that harm is and how serious it is. Our nurse is usually very good at drawing information out and being compassionate when she is dealing with that, in the moment, the counsellors deal with the aftermath, if you like, after the initial situation.”*
*[Interview 31]*

In some cases, there was capacity for a whole school/preventive approach, while in others the overwhelming nature of the work means inferred that this is not able to be implemented and focus was more on the individual:


*“And students are not shocked by that because we talk about suicidal thoughts very openly at the beginning of their life at the school, they all get this workshop on suicidal thought”*
*[Interview 12]*

#### 3.2.3. Subtheme 3 Communication

Discrepancy was also seen regarding communicating information between different people who were engaged with young people, both within and external to the school e.g. parents and family/whānau. The data revealed a discrepancy across schools in terms of how various professionals within the school collaborated and what information was communicated, including between senior management and the health/pastoral care team. There was discrepancy regarding whether physical health professionals, such as nurses, were part of (collectively they might be seen as the wellbeing team) or external to the pastoral care team and how they worked together. There was also variation in terms of who young people see in an ongoing way for intervention. In some schools there was a sense of there being a whole of school approach, and where everyone was responsible to look out and report concerns:


*“Collegiality between the nurses and the other counsellor is absolutely essential”*
*[Interview 22]*


*“We (pastoral care) then try to consult with our nursing team…they don’t want to participate in our approach”*
*[Interview 12]*


*“I set up a bit of a formulaic email, which is good, just to alert to the Dean”*
*[Interview 23]*


*“Grounds-staff emailed me about three weeks ago and says oh there’s this boy…”*
*[Interview 4]*


*“Teachers had noticed things but hadn’t feed it back to us”*
*[Interview 11]*


*“We started an online referral form this year for staff. Um and that has just short circuited all those passing corridor conversations”*
*[Interview 10]*

There appeared to be a general awareness that external agencies were also over-burdened, with some frustration expressed that these services could not accept referrals. This often resulted in pastoral care staff feeling overburdened and under prepared to manage the complex presentations. There was a perception that the response participants received from external agencies varied over time and across different agencies, and that both the trust and communication between themselves and external agencies varied:


*“Yeah, trust us, we’re the professionals, we’re the trained psychologists… um… and we’ve made an assessment and this person isn’t at risk. Whereas we’re the ones that have had lots and lots of in depth conversations with them”*
*[Interview 5]*


*“If we’re not informed of the plan, we can’t help them… um… also vice versa”*
*[Interview 6]*


*“I feel like that work is paying off and there is a widening and there is a more collaborative approach”*
*[Interview 19]*


*“They’re much slower to pick up. But it’s changed, I mean one time I could say really great things…and then other times…there will be a move of staff and then they’ve changed practices and procedures”*
*[Interview 21]*

There was considerable variability in the threshold that was required for providing information to the family/whānau across schools. Some schools had policies about this and gave clear guidance, whilst others did not. Sometimes this guidance applied to acute events rather than an ongoing behavior and stated when it was appropriate to liaise with whānau/family. Likewise, self-harm appeared to warrant a different response from suicidality:


*“Our school policy says…they’ve been self-harming and they’ve just cut themselves…and their family are called and asked to come and take them home”*
*[Interview 13]*


*“I will hold something before I take it to senior management because they’ll always be “let’s call whānau” and sometimes that actually makes a situation bigger… I don’t think there needs to be immediate family involvement*
*[Interview 11]*


*“We do make a distinction between self-harm disclosure and suicidality disclosure…we wouldn’t necessarily disclose to parents about self-harm”*
*[Interview 22]*

### 3.3. Theme 3: Need for Guidelines

The increased burden, and considerable discrepancy across a range of factors relevant to the management of self-harm within schools, was related to a final theme about the need for guidelines. Within this theme, three subthemes emerged, including positive views about the need for guidelines, that there was a general lack of utilization of guidelines in schools generally, and that there needed to be flexibility in the use of guidelines as applied to each unique situation.

#### 3.3.1. Subtheme 1: Positive Views about the Need for Guidelines

Guidelines were perceived as encouraging best practice and providing support in what, as described in Theme 1, was seen as a burdensome and highly stressful situation. Guidelines were also seen as providing a rationale for school policy which provided a basis of agreement and understanding for everyone and promoted a consistency in approach. They provided reassurance and direction when staff needed support with more ambiguous or difficult situations, and to prevent omissions during assessment and management:


*“Situations are not clear cut and guidelines are really helpful to go back and have something that you’re concretely measuring up against to help you work your way through to best practice”*
*[Interview 13]*


*“Young people need to know that wherever they go they’re going to get the same care… And the same questioning and the same assessment and that bothers me that they don’t at this stage.”*
*[Interview 20]*


*“Young people need to know that wherever they go they’re going to get the same care… And the same questioning and the same assessment and that bothers me that they don’t at this stage.” [Interview 20] “That’s where a guideline would come in like as a checkbox like have you thought of this, this, this, this, this. We can actually work your way through because it can be quite an intense thing depending on the time, depending on the situation”*
*[Interview 3]*


*“We realized that that wasn’t really any process for how to deal with it at our school. So, I went online to see what I could track down…the ones for managing traumatic events for schools are like the encyclopedia Britannica. You know it’s too long, too complicated…you’ve got to have it simple so you can use it”*
*[Interview 32]*

#### 3.3.2. Subtheme 2: Lack Utilization of Guidelines

The data revealed that there were no consistently used guidelines; some referred to relevant Ministry of Education guidelines, others referred to their own profession’s guidelines. Other participants were not aware of guidelines existing at all l, or outside of their school. Available guidelines had often been provided after training, rather than as part of an induction process. Overall, what was more common were local school policies and procedures that had been developed or that pastoral care staff intended to create. While an intention to develop policies and procedures was mentioned, this often relied on informal processes. Some participants developed these individually, whilst others had developed these collaboratively with other professionals/colleagues;


*“I know about our schools’ ones, I don’t know about, um, beyond our school”*
*[Interview 13]*


*“I eventually got that training through the Ministry of Health uh Education whereby they had that little wee book”*
*[Interview 22]*


*“I feel the NZAC, my professional body, if I wish to, I’ll go to them for and they will point me in the right direction”*
*[Interview 23]*


*There isn’t like a formal process written down anywhere”*
*[Interview 5]*


*“One of the areas that suffered is that there aren’t protocols here….”*
*[Interview 1]*

#### 3.3.3. Subtheme 3: Need Flexibility

There was a concern that guidelines may not be useful if they did not consider the local community, school, or individual student context. Making the student the center of the guidelines and recommendations was key, with all participants highlighting the importance of a student-centered approach. Given this, there was some understanding that guidelines are just that—guidelines that do allow for flexibility and that this was important for all involved:


*“I think really it’s about having the student at the center of it really and I sometimes think that that gets a bit lost when it comes to procedures and policies”*
*[Interview 18]*


*“I always feel that it is a case-by-case, I really feel that. So, I don’t know how I would go with protocols but they are as you say they guidelines”*
*[Interview 14]*


*“[guidelines] would be really useful because then it becomes a bottom line and then the school can adjust to their school needs”*
*[Interview 26]*

## 4. Discussion

This research engaged 26 school pastoral care staff, providing them with an opportunity to describe their experiences managing students who presented with self-harm. Three key themes arose: The burden of caring for students engaging in self-harm without support; the discrepancies in the approaches between individuals and schools; and the need for guidelines to shape and support practice.

The theme of ‘burden’ captured how excessive many pastoral care staff experienced their workloads and responsibilities. Many pastoral care staff felt that they were not adequately resourced to manage the increased prevalence of mental health issues nor had they had specific training related to these issues. While pastoral care staff tended to express a desire to have an impact on the school culture and engage in preventative and proactive work, this was often unattainable due to excessive demand and under-resourcing. As such, preventative work was often passed over due to the need for acute risk management.

The theme of ‘discrepancy’ described the inconsistencies that emerged from the data with regard to the training in self-harm management that was received, and procedures for managing students presenting with self-harm and communication (or lack thereof) between those involved in students’ pastoral care. This indicated that there is not a cohesive ‘best practice’ and that the response to someone who presents having engaged in self-harm is largely at the discretion of the school pastoral care staff and other school processes. Many school guidance counsellors lamented this reality and reported frustration with their current systems.

Finally, there was a belief that guidelines may help mitigate these discrepancies by making recommendations for best practice, both for school pastoral care staff and wider school staff. Guidelines would allow school pastoral care staff to understand their responsibilities, help prevent omissions in practice, and note when circumstances may exceed their abilities. Indeed, overwhelmingly, school guidance counsellors supported the development of consistent guidelines for managing self-harm in and across schools, and many had taken it upon themselves to draft informal policies and procedures specifically for their school. However, school guidance counsellors also expressed a need for flexibility when applying guidelines, ensuring it was appropriate for their school context and could be tailored to the student(s) presenting for care.

### 4.1. How This Relates to Other Studies/Previous Literature

Participants reported an increase in the incidence of self-harm behaviors, and this appears to be consistent with the New Zealand experience [11]. Participant responses reflected global trends of anxiety about how best to respond, inconsistencies in responding, and the desire for guidelines to support practice [20,28,29,30,31,32] and as such, adds to the growing body of literature supporting the needs for guidelines to support both staff, and students who self-harm. This study will also contribute to future research which will lead to the development of guidelines for schools in New Zealand using a Delphi process (to be completed in 2019/2020).

### 4.2. Limitations

There are several limitations of this study. Firstly, the geographical location of participants may limit the generalizability of the results. Most participants were from New Zealand’s largest metropolitan center which offers multiple mental health services. This is reflective of few other centers in New Zealand, and therefore the demands and the access to other agencies may be different to those in smaller or rural centers.

The study has also only explored experience from the perspective of school counsellors, and not those of other school staff, parents, or the students who should be central to our understanding. Given this, it would be important that the development of guidelines that may result from this work include a wider group, including other mental health professionals; those with expertise managing self-harm, students and their family/whānau.

Whilst the study attempted to purposefully recruit Māori and Pasifika participants, this had limited success and therefore the ability to generalize these findings to Te Ao Māori Kura (schools) and within the Pasifika communities; therefore, findings need to be considered with caution. Of note, the Harakeke Model emphasizes that there is no individual child but that the child is part of a family/whānau that needs to be supported. Given this, separate guidelines may need to be considered for Māori students.

Finally, the interpretation of the interviews has occurred through the lens of the research team and has not included interpretation from the participants. Given this, another team may interpret these findings differently.

### 4.3. Implications for Research and Practice

Findings from this study indicate a desire and need for guidelines to support school staff in managing self-harming young people. Participants highlighted that these guidelines would need clear direction on how to manage self-harm, would help delineate roles, and would place the student at the center of the guidelines. Given no such guidelines currently exist in New Zealand to manage self-harm in schools, it is recommended that future research builds on this study and works towards developing these guidelines. Considering that practitioners often emphasis that young people are central, it would be considered best practice to include young people in the design of these guidelines.

## 5. Conclusions

Overall, this research further highlights the need to support pastoral care within schools. School pastoral care staff, alongside nurses, other school staff, and senior management are at the forefront of managing youth mental health. We need to ensure all school personnel have the tools and resources to facilitate delivery of evidence informed interventions for managing self-harm, especially in a school context where there may be an impact on vulnerable others. This is a necessary step for addressing New Zealand’s poor mental health, especially amongst adolescents. As such, school staff have indicated a desire to have guidelines that they would refer to when supporting a young person who engages in self-harm. Guidelines would help advocate for increased support, be it increased staff to manage workload, as well as establishing expectations within the school community and the parent body. Overall, the demand on school pastoral care staff is unlikely to radically decrease in the near future. Ensuring that those on the forefront can adequately manage students that engage in self-harm is vital.

## Figures and Tables

**Figure 1 ijerph-16-04844-f001:**
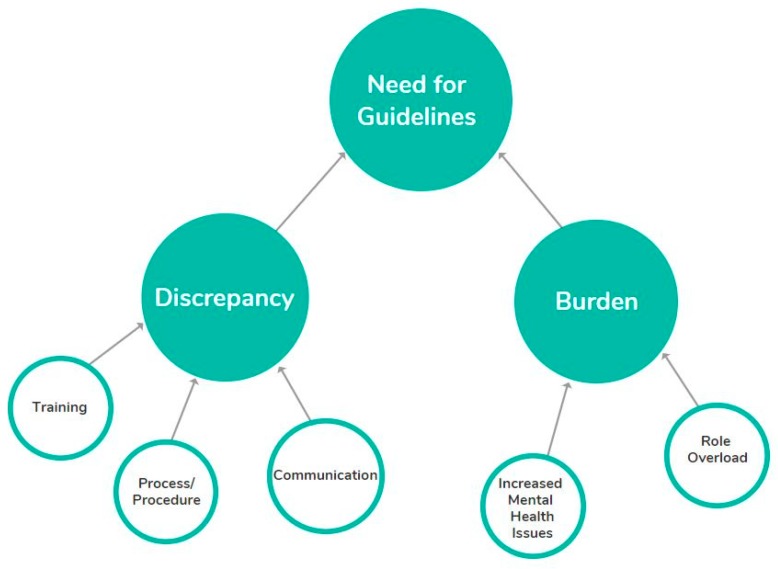
Themes in relation to school guidance counsellors managing students who engage in self-harm.

**Table 1 ijerph-16-04844-t001:** Participant and school characteristics.

Interview	Decile	School Type	Location	Ethnicity (Participant)	Profession	School Population
1	9	State	Central Auckland	NZ European/Pākehā	School Guidance Counsellor	2000+
2	9	State	Central Auckland	Asian	School Guidance Counsellor	2000+
3	9	State	Central Auckland	NZ European/Pākehā	School Guidance Counsellor	2000+
4	9	State	Central Auckland	Samoan	School Guidance Counsellor	2000+
5	10	State	North Auckland	NZ European/Pākehā	School Guidance Counsellor	1500–1999
6	10	State	North Auckland	NZ European/Pākehā	School Guidance Counsellor	1500–1999
7	7	State	Central Auckland	NZ European/Pākehā	School Guidance Counsellor	2000+
8	7	State	Central Auckland	NZ European/Pākehā	School Guidance Counsellor	2000+
9	7	State	Central Auckland	NZ European/Pākehā	School Guidance Counsellor	2000+
10	9	State integrated	North Auckland	NZ European/Pākehā	School Guidance Counsellor	2000+
11	8	State	Central Auckland	NZ European/Pākehā	School Guidance Counsellor	1000–1499
12	4	State	Central Auckland	Dutch	School Guidance Counsellor	1500–1999
13	4	State	Central Auckland	NZ European/Pākehā	School Guidance Counsellor	1500–1999
14	2	State integrated	Central Auckland	NZ European/Pākehā	School Guidance Counsellor	Less than 500
15	4	Private	Central Auckland	NZ European/Pākehā	School Guidance Counsellor	500–999
17	4	Private	Central Auckland	NZ European/Pākehā	Chaplain	500–999
18	10	Private	Central Auckland	NZ European/Pākehā	School Guidance Counsellor	1000–1499
19	4	State	East Auckland	NZ European/Pākehā	School Guidance Counsellor	1000–1499
20	3	State	Central Auckland	NZ European/Pākehā	School Guidance Counsellor	1000–1499
21	3	State	Central Auckland	NZ European/Pākehā	Social Worker	1000–1499
22	10	Private	North Auckland	NZ European/Pākehā	School Guidance Counsellor	1500–1999
23	1	State integrated	South Auckland	NZ European/Pākehā	School Guidance Counsellor	500–999
26	10	Private	South Auckland	NZ European/Pākehā	School Guidance Counsellor	1000–1499
30	8	State	West Auckland	NZ European/Pākehā	School Guidance Counsellor	1000–1499
31	5	State	Northland	English	School Guidance Counsellor	1000–1499
32	5	State	Northland	NZ European/Pākehā	School Guidance Counsellor	1000–1499

**Table 2 ijerph-16-04844-t002:** Themes and subthemes that arose from interviews.

**Need for Guidelines**	
Positive views about the need for guidelines	Encourages best practice
	Provides support in stressful situations
	Provides a rationale for school policy
Lack utilization of guidelines	Believed guidelines did not exist/not accessible/not relevant to New Zealand context
	Relied on informal policies
Need flexibility	Understand individual context
**Burden—Excessive Workload and Responsibility**	
Increased Mental Health Issues	Increased prevalence of suicidal ideation, self-harm, anxiety, depression
	Increase in potential drivers of mental health issues, including self-harm
Role Overload	Change in role responsibility
	Lack of resourcing
**Discrepancy—Differences in the Way That Self-Harm is Managed between Schools**	
Training	Lack of training on self-harm/suicidal ideation during degree
	Relied on professional development, mentoring, colleagues, supervision
	Educating wider staff members
	Educating parents
Process/Procedure	Risk assessment
	Confidentiality
	Referral
	Management strategies
	Preventative/proactive interventions
Communication	Internal—between pastoral care staff and other school staff
	External—between pastoral care staff and external agencies
	External—between school guidance counsellors and whānau

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
