# Peer review of "Understanding the Experience and Needs of School Counsellors When Working with Young People Who Engage in Self-Harm"

_ijerph, 2019, doi:10.3390/ijerph16234844_

Round 1

Reviewer 1 Report

This is an important study, which I believe will be of huge interest to the readership of IJERPH. It is particularly crucial given high rates of self-harm and suicide.

The manuscript is brief and well written. Although the introduction is well written, the literature can be faulted for lack of depth and breath of recent empirical studies. Appropriate research methods were applied. Conclusion was based on findings on the study.

Author Response

We thank the reviewer for their favourable review of the paper and have taken on board the feedback regarding the introduction.  Subsequent to this we have reviewed the introduction and added additional recent empirical literature.

Reviewer 2 Report

Line 16: Change “increasingly” to “increasing”

Self-harm rates are increasingly globally and demand for supporting, treating and

Line 30: Delete “the”

Within the New Zealand the rate of self-harm [9] parallels the worldwide trend and the rate of

Line 37 & 38: Revise the highlighted statement to increase clarity and flow of thought.

Therefore, the school environment, where young people spend much of their time and its

dominant in this stage of psychosocial and emotional development [15-19] is an important context

Line 46: Delete the word “school”.

management of school students who engage in self-harm in order to support the young person and

Line 52. Delete the “s” in aims and change “were” to “was”.

The aims of this project were to understand the perspective of the key school personnel in New

Revise as follows: The aim of this project was to understand…

Line 58: The type of qualitative research design used was not identified (I.e. phenomenological approach).

Author Response

Thank-you for your valuable feedback on the paper. We have amended according to your points below.  Of particular note are the following additions to the paper:

line 38 and 39 were revised to the following: Young people spend much of their time in the school environment and it is a key influencer of psychosocial and emotional development [15-19].  Given this, is an important context for prevention and intervention of self-harm. 

We have indicated that the study was a thematic analysis as did not follow a specific qualitative analysis paradigm such as phenomenology.

Reviewer 3 Report

This is an interesting paper which explores the experience of self-harm from the point of view of pastoral care staff.

I think this is a really relevant study and the authors have gathered a wealth of data.

I have included some comments/questions that I hope the answers to which will add clarity to the ms.

I wondered if there were any themes which emerged within schools?

Other comments/ questions as they arose through the ms.

Abstract- seems to stop abruptly after results. Needs a sentence as to why this research is important/ what do the themes mean? Line 27- sentence regarding suicidal intent. I find this sentence confusing. Line 30 “Within the New Zealand” delete ‘the’ Line 31 ‘OECD country’ – define Line 36 ‘whānau’- font seems to change Line 38 dominant – should this be dominance? Line 39 ‘of self-harm’ – in self-harm? Line 48 I think a little more detail is needed here as this is the justification for the research Line 98 “there organised” – then organised? Line 100- what % of interviews were double coded? What was level of interrater reliability? Table 1- table feels more descriptive of school than participant characteristics: Could participant characteristics such as age and gender be included Is it possible to indicate which of the 16 schools each came from? Table 2. for continuity with the rest of the text, I think burden should be the first row. Line 128 couple of tense changes within this paragraph. Line 146- this feels like a really good example for the text in Line in line 149; line 156 feels like a good example for the text in Line 141. Would swapping the quotes in 146 and 156 work? Line 164- The sentence in regarding educating teachers etc feels out of place here. Moving it to line 187 with the other text on educating others would help the flow of the section. Line 194 “keep a running role of people” I think this should be “roll” as in roll call? Line 266. I read this quote in a different context to this. As it refers to a referral system within the school, rather than referring to another service. Would this be more appropriate under subtheme 3? Line 283 “if something”- is this correct? Line 290- this would make a great example of Subtheme 2: Role overload Line 418- how does the study relate the existent literature regarding young people/ schools? Headings are inconsistent in points and formatting goes a bit awry in places too (e.g. line 358)

I hope that the authors find these comments helpful in revising their manuscript.

Author Response

Dear reviewer,

Thank-you for your valuable feedback. We have really valued your input and insights and amended as per below:

I wondered if there were any themes which emerged within schools? New Zealand has a small population as so there was only two occasions where staff from the same school were interviewed.  Given this, we did not feel that this would b adequate to draw conclusions from.  It would certainly be very interesting to explore in the future, particularly looking at schools for Maori vs. schools for non-Maori. The abstract seems to stop abruptly after results. Needs a sentence as to why this research is important/ what do the themes mean? We agree!  We have added the following to the abstract: " This research, therefore, demonstrated a need for guidelines to support school staff to provide support around decision making and response to self-harm in the school environment." Line 27- sentence regarding suicidal intent. I find this sentence confusing. We have reviewed this sentence to read: "(self-inflicted injury of self-poisoning irrespective of motivation or the individual’s intent to die or not [1])" and hope this provides more clarity. Line 30 “Within the New Zealand” delete ‘the’ Completed Line 31 ‘OECD country’ – define We have defined OECD immediately prior to the abbreviation Line 36 ‘whānau’- font seems to change It did - we have changed the font and searched for other incidences where this may occur and have amended these. Line 38 dominant – should this be dominance? Correct, this has been amended Line 39 ‘of self-harm’ – in self-harm? This has been corrected Line 48 I think a little more detail is needed here as this is the justification for the research This has been amended to read: "Schools are a logical and accepted setting for suicide prevention activities, and there is growing evidence to support this [26,27]. It is also critical that schools are swift and effective in their management of students who engage in self-harm in order to support the young person and ensure the risk of contagion is minimized. School staff, despite often being the first port of call for at-risk students, report feeling limited in their capacity to respond appropriately to these students [20,28,29]. And there is a lack of clarity around how well schools and their staff, including those specifically responsible for the mental health and wellbeing of students, are resourced and equipped to engage in this kind of work. This can sometimes result in confusion about how to respond, unhelpful responses to self-harm [30], and a lack of clear plan to support the student and the school [20].

Recently, there has been a focus on supporting staff and students through position papers and guidelines outlining when and how to respond to self-harm in the school environment [30-32].  However, these do not adequately address the unique cultural challenge of working in New Zealand.  The aim of this project was to understand the perspective of the key school personnel in New Zealand responsible for the mental health and wellbeing of students. We wanted to know what their experience was of managing students who engaged in self-harm, gain insight into how these students were managed within the school, including the use of guidelines, protocols and procedures, as well as understand how well resourced staff were to manage these studies.  Finally, we sought to understand if guidelines were required, or would be welcomed."

Line 98 “there organised” – then organised? This has been corrected Line 100- what % of interviews were double coded? What was level of interrater reliability? We have provided details around the number of interviews double coded, but not on the interrater reliability as this was an iterative learning process designed to teach junior staff about coding as well as building our coding frame. Table 1- table feels more descriptive of school than participant characteristics: Could participant characteristics such as age and gender be included. Is it possible to indicate which of the 16 schools each came from? We agree and have opted to keep it like this due to the small nature of NZ making participants potentially identifiable if we provide demographic characteristics linked to schools.  We have included the demographic characteristics in the text but it does not link to the tables. Table 2. for continuity with the rest of the text, I think burden should be the first row. We have amended this as per the suggestion Line 128 couple of tense changes within this paragraph. This has been reviewed and consistency of tense ensured. Line 146- this feels like a really good example for the text in Line in line 149; Agreed and has been amended to reflect this. line 156 feels like a good example for the text in Line 141. We have amended this as per the suggestion Would swapping the quotes in 146 and 156 work? After completing the above two points it no longer feels as though swapping these would be as beneficial.  We have therefore left it as amended. Line 164- The sentence in regarding educating teachers etc feels out of place here. Moving it to line 187 with the other text on educating others would help the flow of the section. Thank-you for this suggestion. After discussion with the senior authors we have decided to leave this as it is currently written as it is part of setting the theme up (but recognise it could go either way). Line 194 “keep a running role of people” I think this should be “roll” as in roll call? We have amended this as per the suggestion Line 266. I read this quote in a different context to this. As it refers to a referral system within the school, rather than referring to another service. Would this be more appropriate under subtheme 3? We do agree with this and has amended accordingly Line 283 “if something”- is this correct? This has been amended as this was a typo. Line 290- this would make a great example of Subtheme 2: Role overload Agreed, and has been moved into this sectoin. Line 418- how does the study relate the existent literature regarding young people/ schools? This paragraph has been edited to link to the existing literature.  Headings are inconsistent in points and formatting goes a bit awry in places too (e.g. line 358) We have reviewed the manuscript for consistency and amended accordingly.

Thank-you again for your feedback - it has been invaluable in refining our manuscript.